# An Enhanced YOLOv5 Model for Greenhouse Cucumber Fruit Recognition Based on Color Space Features

Ning Wang [1], Tingting Qian [2,3,4,*], Juan Yang [2,3,4], Linyi Li [2,3,4], Yingyu Zhang [2,3,4], Xiuguo Zheng [2,3,4], Yeying Xu [2,3,4], Hanqing Zhao [3,4,5] and Jingyin Zhao [3,4,6,*]

1. College of Information Technology, Shanghai Ocean University, Shanghai 201306, China
2. Institute of Agricultural Science and Technology Information, Shanghai Academy of Agricultural Sciences, Shanghai 201403, China
3. Shanghai Engineering Research Center of Information Technology in Agriculture, Shanghai 201403, China
4. Key Laboratory of Intelligent Agricultural Technology (Changjiang Delta), Ministry of Agriculture and Rural Affairs, Shanghai 201403, China
5. College of Information and Management Science, Henan Agricultural University, Zhengzhou 450046, China
6. Shanghai Association of Senior Scientists and Technicians, Shanghai 200070, China
* Correspondence: qiantingting@saas.sh.cn (T.Q.); zjy@saas.sh.cn (J.Z.); Tel.: +86-150-0075-3513 (T.Q.); +86-189-1816-2078 (J.Z.)

**Abstract:** The identification of cucumber fruit is an essential procedure in automated harvesting in greenhouses. In order to enhance the identification ability of object detection models for cucumber fruit harvesting, an extended RGB image dataset ($n = 801$) with 3943 positive and negative labels was constructed. Firstly, twelve channels in four color spaces (*RGB*, *YCbCr*, *HIS*, *La\*b\**) were compared through the ReliefF method to choose the channel with the highest weight. Secondly, the RGB image dataset was converted to the pseudo-color dataset of the chosen channel (*Cr* channel) to pre-train the YOLOv5s model before formal training using the RGB image dataset. Based on this method, the YOLOv5s model was enhanced by the *Cr* channel. The experimental results show that the cucumber fruit recognition precision of the enhanced YOLOv5s model was increased from 83.7% to 85.19%. Compared with the original YOLOv5s model, the average values of *AP*, *F1*, recall rate, and *mAP* were increased by 8.03%, 7%, 8.7%, and 8%, respectively. In order to verify the applicability of the pre-training method, ablation experiments were conducted on SSD, Faster R-CNN, and four YOLOv5 versions (s, l, m, x), resulting in the accuracy increasing by 1.51%, 3.09%, 1.49%, 0.63%, 3.15%, and 2.43%, respectively. The results of this study indicate that the *Cr* channel pre-training method is promising in enhancing cucumber fruit detection in a near-color background.

**Keywords:** deep learning; color space; ReliefF characteristic analysis; near color recognition

## 1. Introduction

Cucumber is one of the most popular greenhouse vegetables worldwide, which is widely cultivated and comparatively productive. However, the workload during harvest is considerable, which also increases the labor cost [1]. Therefore, it is necessary to develop intelligent and automated cucumber-picking equipment to reduce the labor input. To date, using automatic identification to assist automatic picking has been realized in tomato [2], citrus [3], litchi [4], apple [5], strawberry [6], mango [7], etc. The colors of these ripe fruits are greatly different from the surrounding environment and are easy to identify. However, the color of harvestable cucumber fruit is green, which is similar to its surrounding environment. In addition, occlusion between fruits and leaves poses a challenge to the identification of cucumber fruits using an object detection model under natural conditions. Therefore, it is of great significance to improve the recognition precision of the model in the detection of cucumber fruits in complex near-color backgrounds.

Compared with the industrial field, it is more difficult to realize recognition based on machine vision in agricultural cultivation processes, mainly for the following reasons: (1) The growth environment of the object to be detected is more complicated due to the messy background and inconspicuous object; (2) various light conditions and camera orientations affect the consistency of image parameters such as saturation and brightness; and (3) the occlusion issues, together with the shape and color of fruits such as cucumbers, raise the difficulty in fruit recognition.

In previous studies, machine learning has been used to manually extract features and train classifiers from shallow features such as shape, surface texture and color, so as to detect and identify fruits. Malik et al. [8] used the combined threshold method to remove the background in the *HSV* color space to segment mature red tomato fruit. However, in a near-color background, this method could filter out the green immature tomatoes along with the leaves, resulting in missing detections. To solve this problem, Liu et al. [9] proposed a method based on color and shape features to detect immature green apples, using super-pixel segmentation and the Hough transform to perform ellipse fitting on apple fruits. However, the Hough transform is no longer applicable for cylindrical fruits such as cucumbers. In this case, Li et al. [10] converted *RGB* images to other color space images to classify and segment the cucumber fruit. As previously stated, most of the recognition methods based on machine learning enhanced the extraction of image color features from the color space conversion [11,12]. However, they achieved good accuracy at the cost of efficiency. For instance, feature selection was conducted manually, which is a time-consuming and heuristic method that highly relies on experience.

Convolutional neural network based on deep learning have a strong ability to express and extract features of images. They make use of the image itself to promote the self-optimization learning of the relationship between features and expression, which is fast and accurate. In order to improve the accuracy of deep learning object detection models, researchers are currently thinking about combining the deep learning method with color space conversion [13]. Liu et al. [14] converted an original potato *RGB* image to the *HSL*, *HSV*, *Lab*, *XYZ*, and *YCrCb* color spaces and then created Mask R-CNN models for each color space to detect wilt plaque on leaves. However, the Mask R-CNN model requires datasets with pixel-level annotation, which are difficult to construct. However, cucumber fruit recognition only requires the positioning information. Alli et al. [15] expanded the color space for training images by generating composite images with modified color value distributions, improving deep learning neural networks' ability to identify cassava diseases in low-quality images. In addition, there were methods in the industrial field that combined color space conversion and deep learning to detect and identify specific objects [16,17]. Though these methods are good at recognizing objects that are different colors from their surroundings, it is difficult to recognize objects in near-color backgrounds, such as harvestable cucumber fruits.

YOLO series models are currently utilized extensively in agriculture for tasks including fruit detection, disease detection [18–20], and pest [21,22] and grass identification [23–25]. As for fruit detection, they are more commonly applied in greenhouses for tomato detection [26], rather than cucumber detection. The aim of this study was to enhance the YOLOv5s model using color information to improve the accuracy of cucumber fruit recognition. The outcome of this paper can serve as a reference for the task of fruit recognition in near-color backgrounds.

## 2. Materials and Methods

### 2.1. Acquisition and Processing of Datasets

The iPhone13 pro rear camera was used to collect cucumber fruit images in the glass greenhouse of Chongming (31°34′ N, 121°41′ E) Base of the National Engineering Research Center of Protected Agriculture on 18 January 2022. The camera parameters were consistent during the collection process. The cucumber variety was Delta star RZ F1 hybrid cucumber (Rijk Zwaan Company, De Lier, The Netherlands). There were 720 cucumber plants grown

under sufficient water and fertilizer, not infected with pests and diseases. The samples (*n* = 200) were randomly selected from the population. A total of 438 original images were collected. For each sample, 2 or 3 images of fruits were taken at random orientations and exposures (Table 1 and Figure 1). Occlusions (Figure 2) were also considered to increase the randomness and diversity of the training samples. The obtained images were marked using LabelImg. The label information included the category and position coordinates of objects and was used for the training of the neural network.

**Table 1.** Classifications of images (*n* = 801) and labels (*n* = 3943).

| Images | Exposure | | Orientation | | |
|---|---|---|---|---|---|
| | **Under-exposed** 336 | **Over-exposed** 102 | **Upward view** 102 | **Top view** 109 | **Side view** 227 |
| **Labels** | **Positive** | | **Negative** | | |
| | **Intact fruits** 1841 | **Occluded fruits** 326 | **Segmented fruits** 170 | **Female flowers** 237 | **Leaves and stems** 1515 |

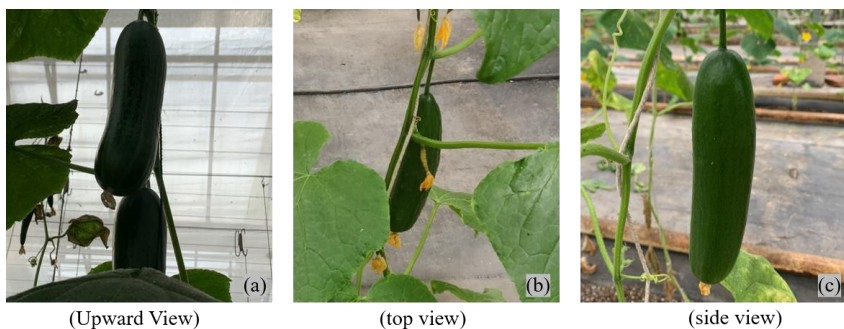

(Upward View)　　　　(top view)　　　　(side view)

**Figure 1.** Upward view (**a**), top view (**b**), and side view (**c**) of cucumbers obtained using different camera orientations.

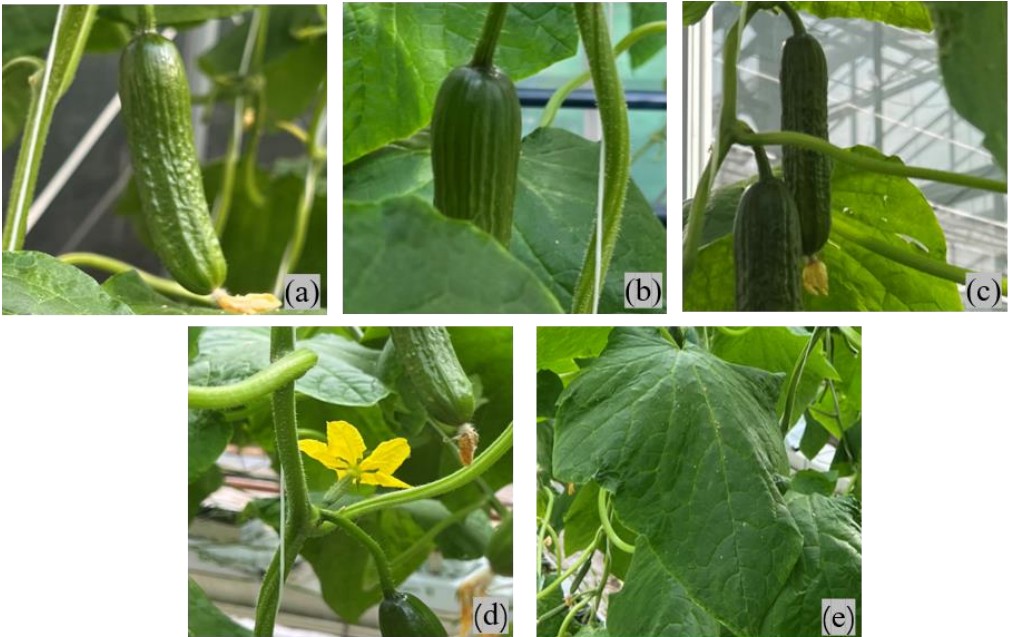

**Figure 2.** Positive labels (**a**–**c**) with different degrees of occlusion and negative labels (**d**,**e**). For each category: (**a**) intact cucumber; (**b**) partial cucumber occluded by leaves; (**c**) cucumber sliced by stems or petioles; (**d**) female flower; and (**e**) leaves and stems.

Cucumber fruits were divided into three categories, including intact (Figure 2a), partially occluded (Figure 2b), and partially sliced (Figure 2c) cucumbers which were set as positive labels. In addition, female flowers (Figure 2d) and leaves and stems (Figure 2e) were set as negative labels, which effectively reduced the recognition errors in the near-color environment. In order to improve the complexity of the dataset, we further applied Gaussian noise (Figure 3a), random noise (Figure 3b), and salt and pepper noise (Figure 3c) to extend the dataset from 438 to 801. After data extension, the number of positive labels (a–c) was 2191, and the number of negative labels (d, e) was 1752 (Table 1). Each image contained 2.74 cucumber labels on average (n = 2191/801).

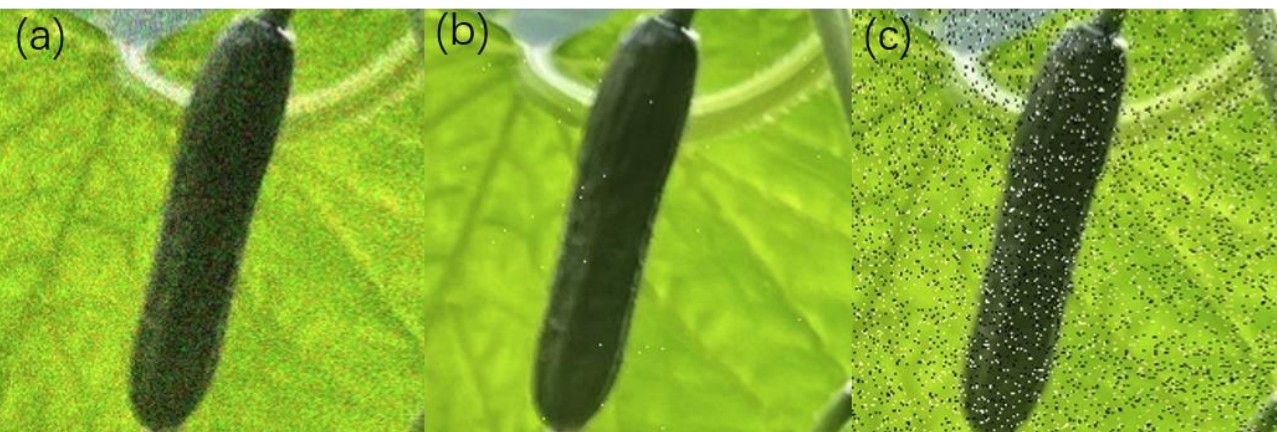

**Figure 3.** Different types of noise applied to cucumber images: (**a**) Gaussian noise; (**b**) random noise; and (**c**) salt and pepper noise.

The dataset was divided into a training set and a validation set in the ratio of 8:2. An additional 100 images, collected at the experimental base of Shanghai Jiaotong University (31°7′12″ N, 121°22′48″ E) on 11 May 2021, were used for testing. The label format marked by LabelImg software was VOC, which needed to be converted to the txt format required by the YOLOv5 model.

### 2.2. Color Space Conversion and Weight Analysis

Different color spaces can illustrate color information from different perspectives. By converting the *RGB* image to another color space, various representations of cucumber fruits and leaves were obtained in various color spaces.

#### 2.2.1. Conversion Principles

The color spaces used in the experiment include *RGB*, *HSI*, *La\*b\** and *YCbCr*, with a total of 12 channels. Pixel values were converted from *RGB* to different color spaces. The conversions of the color space were computed as follows:

*RGB* to *HIS*:

$$\theta = cos^{-1}\left\{\frac{(R-G)+(R-B)}{2\sqrt{(R-G)^2+(R-B)(G-B)}}\right\} \tag{1}$$

$$H = \begin{cases} \theta, & B \le G \\ 360 - \theta, & B > G \end{cases} \tag{2}$$

$$S = 1 - \frac{3 \cdot \min(R, G, B)}{R + G + B} \tag{3}$$

$$I = \frac{R + G + B}{3} \tag{4}$$

*RGB* to *YCbCr*:

$$Y = 0.299R + 0.587G + 0.114B \tag{5}$$

$$Cb = 0.564(B - Y) \tag{6}$$

$$Cr = 0.713(R - Y) \tag{7}$$

*RGB* to *Lab*:

$$L = 0.213 \cdot R + 0.715 \cdot G + 0.072 \cdot B \tag{8}$$

$$a = 0.326 \cdot R - 0.499 \cdot G + 0.173 \cdot B + 128 \tag{9}$$

$$b = 0.122 \cdot R + 0.379 \cdot G - 0.500 \cdot B + 128 \tag{10}$$

### 2.2.2. ReliefF Weight Analysis Method

ReliefF [27] is a feature weight analysis algorithm. Its fundamental tenet is to weight various features in accordance with the correlation between features and the corresponding categories. When the weight is less than a certain threshold, the feature is eliminated. When dealing with discrete problems such as classification, a sample $D$ is randomly selected from the training set each time, and the number of sample extraction is $m$. Then, $k$ nearest neighbor samples $H_j$ of the same type as $D$ and $k$ nearest neighbor samples $M_j(C)$ of different types as $D$ are found. Following this, the weight of each feature is updated according to the Formula (11):

$$W(A) = W(A) - \sum_{j=1}^{k} \frac{diff(A,D,H_j)}{mk} + \sum_{C \notin class(A)} \left[ \frac{p(C)}{1 - p(class(D))} \sum_{j=1}^{k} \frac{diff(A,D,M_j(C))}{mk} \right] \tag{11}$$

*W(A)* is the weight of feature *A*, *p(C)* is the class proportion, *p(class(D))* is the proportion of the randomly chosen sample class, and *diff (A, D1, D2)* is the difference between samples *D1* and *D2* over feature *A* (Equation (12)).

$$diff(A, D_1, D_2) = \begin{cases} 0, & if\ D\ is\ discrete\ and\ D_1[A] = D_2[A] < 0 \\ 1, & if\ D\ is\ discrete\ and\ D_1[A] \neq D_2[A] < 0 \end{cases} \tag{12}$$

In each picture, three $32 \times 32$-pixel fruit areas and two $32 \times 32$-pixel leaf areas were extracted. A total of more than 200,000 pieces of data were calculated by the ReliefF algorithm.

### 2.3. Model Selection and Experimental Environment

#### 2.3.1. YOLOv5

The one-stage YOLO series of deep learning models can detect objects in images quickly and accurately using direct regression [28–30]. Considering that the latest YOLOv5s model has better improvements on Mosaic, CSPDarknet53, Mish and Dropblock [31–34], this study used YOLOv5s version for object detection.

The YOLOv5s model is generally divided into four parts (Figure 4): input, backbone, neck, and prediction. (1) Input: The image data are preprocessed by Mosaic data enhancement, normalized and scaled to the required size of the network, which enhances the detection ability of the model for small objects. Then, the adaptive anchor frame calculation method is used. The initial anchor frame is pre-set for various datasets and detection objects. (2) Backbone: The feature extraction ability of YOLOv5 is enhanced by combining the Focus structure and CSP structure in the main framework of the network. The residual component ensures that the low-dimensional features are not lost in the deep convolution operation. The Mish activation function and Dropblock regularization are used to implement neuron activation and deletion. (3) Neck: In the neck network, FPN and PAN are combined to realize feature fusion and enhance the robustness of the network model. (4) Prediction: CIOU_Loss is used as the border regression loss function, whereas the classification and regression are used to predict the location and category of the object boxes.

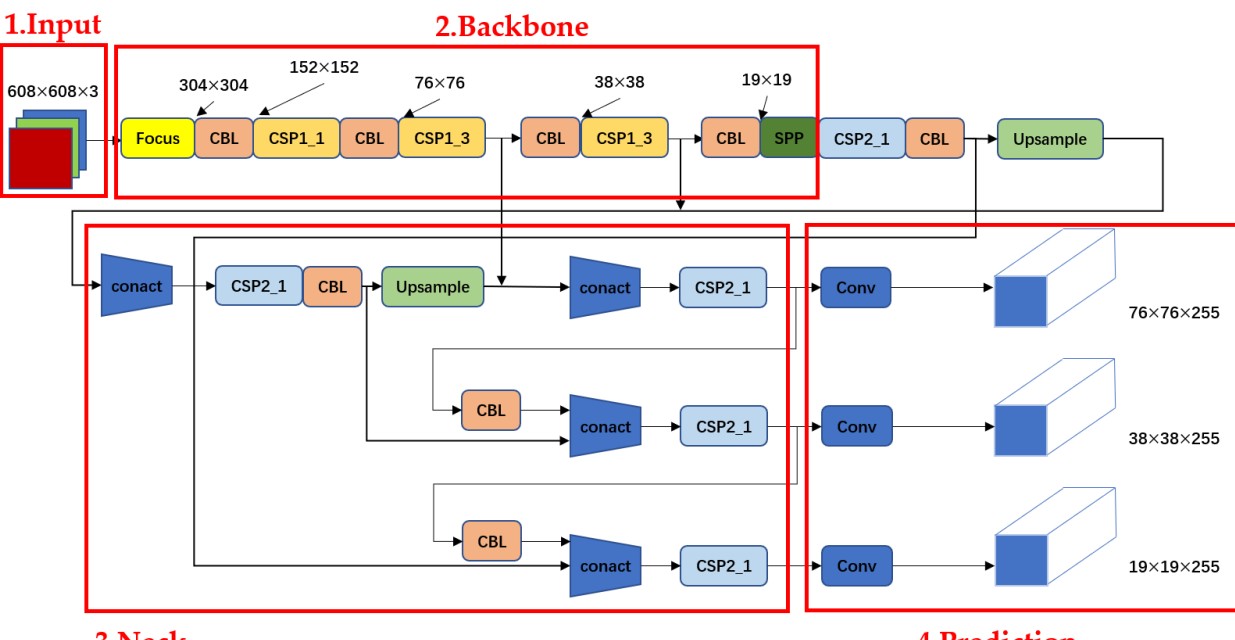

**Figure 4.** The network structure of YOLOv5, which is composed of the input module, backbone network, neck network, and prediction network. (CSP: Cross Stage Partial Network; SPP: Spatial Pyramid Pooling; Conv: Convolutional Layer; Concat: Concatenate Function).

2.3.2. Operating Environment

The above algorithm was run under the following computer configurations: Windows 10 operating system; AMD R9 5950x CPU; NVIDIA RTX3080 graphics card, CUDA 10.1, and CUDNN 7.6.4. toolkit. Focus, CSPDarknet 53. FPN+PAN were used as deep learning model structure. The OpenCV image processing toolkit based on Python was used in the color space conversion.

*2.4. Experimental Process*

A brief flow chart of the experiment is shown in Figure 5.

Four hundred and thirty-eight original images were classified and labeled before conversion to different color spaces in four steps: (1) First, we obtained single channel image data in the *HSI*, *La*b**, and *YCbCr* color spaces. (2) Second, we analyzed the weight of each channel image. (3) Third, we trained the YOLOv5s model with image data from various channels. (4) Fourth, we assessed the detection effect in each channel. Thus, the correlation between the weight of different channels and the enhancement of the recognition ability of the model was proved. After that, the original image dataset was processed by adding noise, blurring, and rotation to extend the dataset to 801. The extended dataset was then used for training and detection after pre-training with the YOLOv5s model using the tested *Cr* channel, and the performance of the model was assessed. Finally, to validate the efficacy of this experimental method in different models, ablation experiments were performed using the SSD, Faster R-CNN, and YOLOv5s/l/m/x models.

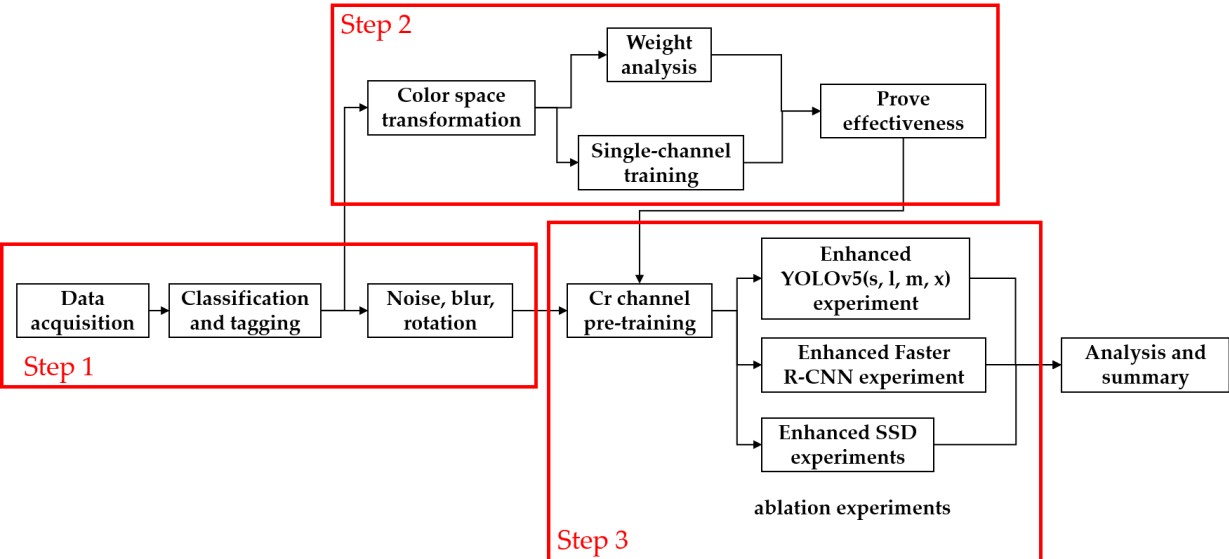

**Figure 5.** A brief flow chart of the experiment: Step 1: Obtain the images and construct dataset. Step 2: Analyze the weight of each channel to choose the channel with the highest weight. Step 3: Use the chosen channel (*Cr* channel) to pre-train the YOLOv5s model before formal training using the *RGB* image dataset. Ablation experiments were conducted on SSD, Faster R-CNN and four YOLOv5 versions (s, l, m, x) to verify the applicability of this method.

### 2.5. Evaluation Indicators

The precision *P* (Equation (13)), recall *R* (Equation (14)), average precision *mAP*:0.5 (Equation (15)), and *F1* (Equation (16)) were used to evaluate network performance. The specific calculations are as follows:

$$P = \frac{TP}{TP + FP} \tag{13}$$

$$R = \frac{TP}{TP + FN} \tag{14}$$

$$F1 = \frac{2 \cdot P \cdot R}{P + R} \tag{15}$$

$$mAP = \frac{\sum_{i=1}^{k} AP_i}{k} \tag{16}$$

where *TP* is the positive sample that is correctly identified as the positive sample; *TN* is the negative sample that is correctly identified as the negative sample; *FP* is the negative sample that is wrongly identified as the positive sample; and *FN* is the positive sample that is wrongly identified as the negative sample.

## 3. Results

### 3.1. Comparative Analysis of Different Color Spaces

The images converted from the *RGB* color space to *HSI*, *YCrCb*, *La\*b\** color spaces are shown in Figure 6. Since the weight analysis process requires a large amount of calculation, 10 random samples were selected from the original images for subsequent single-channel weight analysis for a total of 20 times.

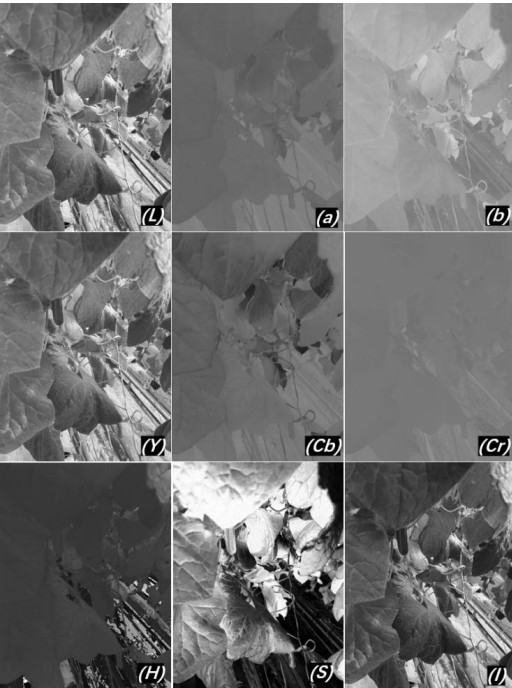

**Figure 6.** The first row is the image in the *La\*b\** color space, with the *L*, *a\** and *b\** channels; the second row is the *YCbCr* color space, with *Y*, *Cb*, *Cr* channels; and the third row is the *HSI* color space, with the *H*, *S* and *I* channels.

The result of the ReliefF weight analysis is shown in Figure 7. Among the 12 channels (*R, G, B, H, S, I, Y, C, b, C, r, L, a\*, b\**), the *Cr* channel from the *YCbCr* color space had the highest weight in identifying cucumber fruits from the background ($p < 0.001$). Therefore, in the *Cr* channel, the difference between the green cucumber fruit and the green background was the greatest.

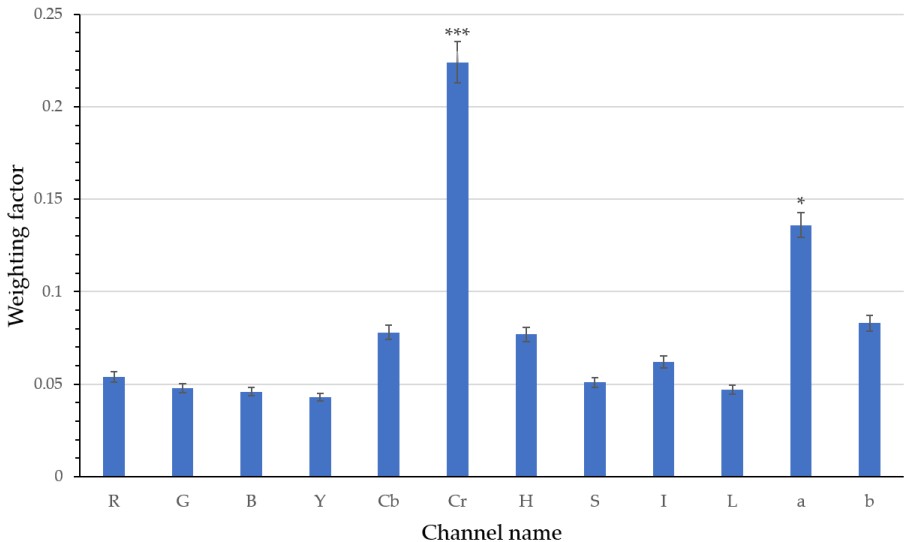

**Figure 7.** Weight analysis results of single channel. Possible differences in the weight between the channels were tested with an ANOVA. Differences between the channels were pairwise compared via Tukey Contrast using the 'glht' function in the 'multicomp' package of RStudio version 1.2.5033. (Significance: *** $p < 0.001$, * $p < 0.05$).

The images of each channel were converted to a pseudo-color dataset to train the YOLOv5s model. The training results are shown in Figure 8. Each single-channel pseudo-

color dataset was learned and trained for 300 rounds, and the *mAP* value of the current test was recorded every 10 rounds. According to Figure 8, the cucumber recognition effectiveness under *Cr* channel has obvious advantages over the other channels, which proves that the weight of the *Cr* channel is positively correlated with the recognition effect of the enhanced model.

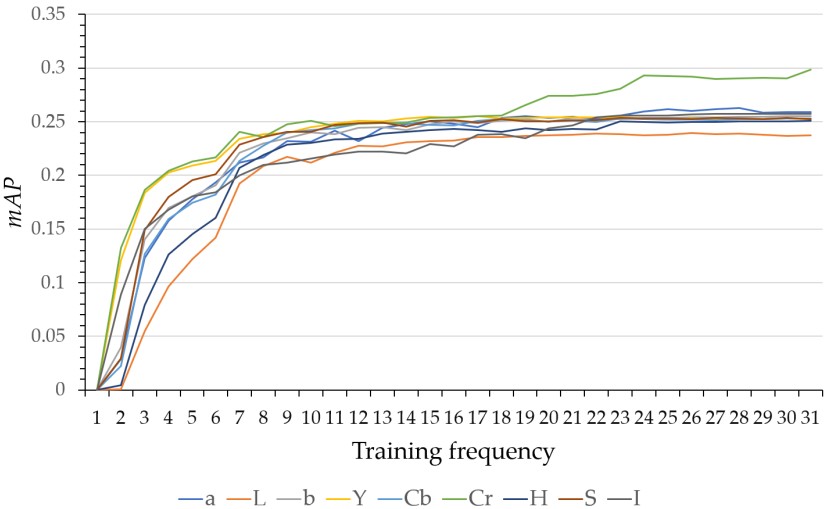

**Figure 8.** Performance results of single channel data with the YOLOv5s model.

### 3.2. Cr Channel Enhances YOLOv5 Recognition Ability

The YOLOv5s model was pre-trained with the pseudo-color image data of the *Cr* channel and then trained on the RGB image dataset. Thus, the ability of the YOLOv5s model to distinguish cucumber fruit from the background in *RGB* images could be enhanced. The recognition ability of the models with and without *Cr* channel pre-training was compared. The F1 score, recognition precision, and *mAP* value are shown in Table 2.

**Table 2.** Comparison of evaluation indexes of enhanced YOLOv5s.

|  | F1 | Precision | Recall | mAP |
|---|---|---|---|---|
| without *Cr* | 0.58 | 83.70% | 44.40% | 50.00% |
| with *Cr* | 0.65 | 85.19% | 53.10% | 58.00% |

All of the evaluation indices were improved. The *F1* score increased from 0.58 to 0.65, the recall rate increased from 44.4% to 53.10%, the *mAP* value increased by 8%, and the precision increased from 83.7% to 85.19%, indicating that using *Cr* single-channel pre-training improves the YOLOv5s object detection model.

### 3.3. Ablation Experiments of Different Object Detection Models

The applicability of the *Cr* channel in enhancing the recognition ability was further confirmed with the SSD, Faster R-CNN, and YOLOv5 (s, l, m, x) models. The evaluation results are as Table 3.

As shown in Table 3, except for the *F1* score in the SSD model and the *mAP* index in the YOLOv5x model remaining unchanged, the evaluation indexes of other models have improved to varying degrees. According to each version of YOLOv5, the precision increased to a maximum of 85.19% under the s version, with an increase of 1.49% over the original version. The frame rates (FPS) for the SSD and Faster R-CNN models were 28.9 and 22.5, respectively. However, the precision of these two models was only about 55%. Moreover, along with the network depth, the *FPS* of YOLOv5 continued to decline, from 53.3 *FPS* in the s version to 14.7 *FPS* in the x version. Because this study did not involve changes in the structure of the YOLOv5 model, the changes in *FPS* before and after improvement were not large.

**Table 3.** Comparison of evaluation indexes of enhanced YOLOv5s model.

|  |  | F1 | Precision | Recall | mAP | FPS |
|---|---|---|---|---|---|---|
| SSD | original | 0.41 | 53.65% | 26.11% | 42% | 28.3 |
|  | enhanced | 0.41 | 55.16% | 26.11% | 42% | 28.9 |
| Faster R-CNN | original | 0.50 | 51.66% | 48.23% | 46% | 21.7 |
|  | enhanced | 0.54 | 54.75% | 53.54% | 51% | 22.5 |
| YOLOv5s | original | 0.63 | 83.70% | 52.21% | 53% | 53.2 |
|  | enhanced | 0.67 | 85.19% | 54.03% | 58% | 53.3 |
| YOLOv5m | original | 0.64 | 80.54% | 52.21% | 53% | 38.6 |
|  | enhanced | 0.65 | 83.69% | 53.10% | 56% | 39.1 |
| YOLOv5l | original | 0.64 | 80.54% | 53.10% | 54% | 24.4 |
|  | enhanced | 0.66 | 81.17% | 55.31% | 57% | 24.5 |
| YOLOv5x | original | 0.61 | 78.95% | 52.21% | 55% | 13.2 |
|  | enhanced | 0.64 | 81.38% | 53.10% | 55% | 14.7 |

### 3.4. Comparison of Detection Effects

The recognition effect of the original YOLOv5s model and the enhanced YOLOv5s model are compared in Figures 9 and 10.

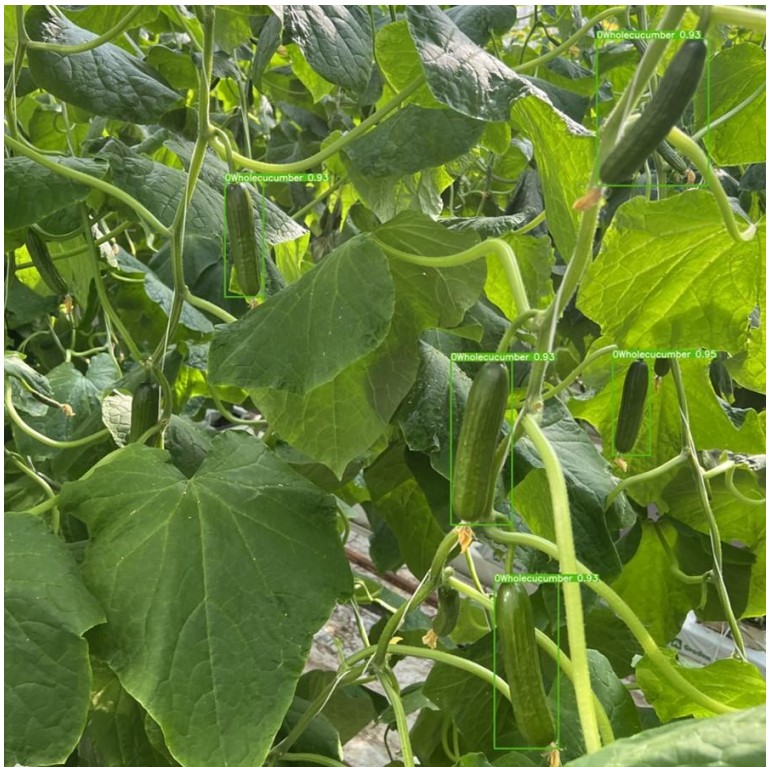

**Figure 9.** Detection effect of the original YOLOv5s models.

The detection results of the original YOLOv5s model are shown in Figure 9, where only relatively intact fruit are identified. The recognition ability for occluded fruits is poor, and small fruits cannot be distinguished. Figure 10 shows the test results using the enhanced YOLOv5s model. Most cucumber fruits are accurately identified, and incomplete fruits can also be accurately classified. The result is significantly better than that of the original model. From the comparison, the enhanced YOLOv5 method has a good recognition ability and high confidence for intact cucumber fruit, partially occluded cucumber fruit, and sliced cucumber fruit in a relatively complex background. At the same time, labeling the negative samples of leaves and stems as the background can greatly avoid identifying green leaves or stems as cucumber fruits and improve the recognition accuracy.

For prediction, 100 images of cucumber fruits were divided into 10 groups, each consisting of 10 images. The comparison results of the number of fruits in each image group counted by the enhanced YOLOv5s model and manual statistics can be found in Table 4.

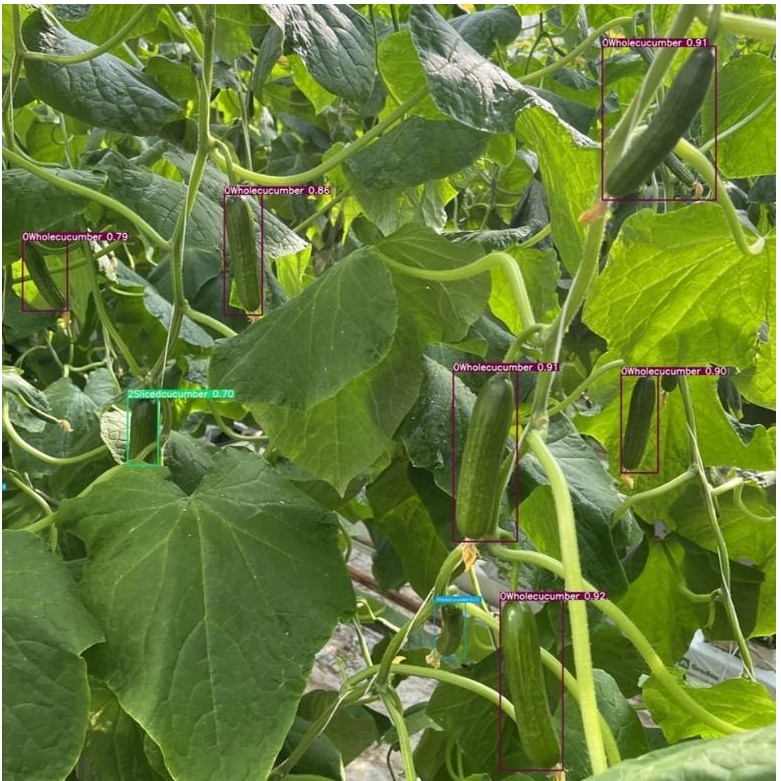

**Figure 10.** Detection effect of the enhanced YOLOv5s model.

**Table 4.** Statistical table of prediction results.

|  | **Manual Statistics** | | **Model Detected** | | **P** |
|---|---|---|---|---|---|
|  | **Complete Fruit** | **Incomplete Fruit** | **Complete Fruit** | **Incomplete Fruit** |  |
| Group 1 | 35 | 11 | 28 | 9 | 80.4% |
| Group 2 | 25 | 9 | 21 | 5 | 76.4% |
| Group 3 | 33 | 12 | 27 | 8 | 77.8% |
| Group 4 | 24 | 10 | 19 | 7 | 76.5% |
| Group 5 | 25 | 11 | 22 | 7 | 80.6% |
| Group 6 | 32 | 6 | 29 | 3 | 84.2% |
| Group 7 | 34 | 5 | 31 | 2 | 84.6% |
| Group 8 | 35 | 9 | 29 | 6 | 79.5% |
| Group 9 | 32 | 7 | 28 | 5 | 84.6% |
| Group 10 | 26 | 11 | 21 | 7 | 78.4% |

According to Table 4, both the intact fruit in the image and the partially occluded fruit were correctly identified by the enhanced YOLOv5s model, with an average recognition precision of 80.3%. The missing detections are due to cucumbers with few pixels and those that are far from the lens, which have no influence in automated harvesting.

## 4. Discussion

The enhanced YOLOv5s model used in this study improved its capacity to recognize cucumber fruit in a near-color background, which can promote the development of automated cucumber harvesting robots and reduce the labor cost. This section discusses the results of this study, possible limitations, and suggestions for the future studies.

*4.1. Result Analysis*

Although most of the previous studies that used machine learning to extract the object from the near-color background achieved high accuracy, the methods were time-consuming and highly dependent on the researchers' experience not only in RGB images, but also in spectral images [35,36]. In order to eliminate reliance on professional experience, this study weighted the feature information through the ReliefF method to choose the color channel based on the difference between cucumber fruit and the near-color background. Then, the deep learning model was trained by channel with the highest weight to enhance the recognition ability. Similar research was conducted by Li et al. [37]. They improved the accuracy of green apple detection to 90.12% using the *Cr* channel. However, the *Cr* channel was used after the application of the YOLOv3 model. Therefore, their method still relies on the detection ability of the YOLOv3 model instead of enhancing the detection ability of the model for the ground truth.

Compared to the YOLOv3 and YOLOv4, some of the improvements in YOLOv5 model play a necessary role in this study. At every epoch, the YOLOv5 adaptive anchor frame computation algorithm can adjust the size of the anchor frame to match the cucumber. However, in YOLOv3 and YOLOv4, the size of the anchor frame is a constant. Besides, the YOLOv5's unique structure (Focus) makes it easier to extract the feature of cucumbers. And lastly, due to the loss function (CIOU_Loss & DIOU_nms) for bounding box used in YOLOv5, the parallel overlapping cucumber fruits can be distinguished easier.

As shown in Table 3, the precision of the YOLOv5l and YOLOv5x versions was lower than that of the YOLOv5m version. It is because the depth of the YOLOv5 model network in the l and x versions is larger than in the m and s versions. Furthermore, compared with the lightweight and miniaturized network of the m version, it is easier for the l and x versions to lose feature information, resulting in a decline in accuracy. In this case, the performance of object detection does not necessarily improve with the depth of the network. For datasets with more low-dimensional features, lightweight and small networks may have better results.

The average recognition precision (Figure 8) from various color space training models under the same circumstances demonstrates that the color information of the *Cr* channel enabled cucumber fruits and leaves to be distinguished more easily than with the other color spaces. This was consistent with the outcomes of the weight analysis process (Figure 7). The result of this paper is also in line with previous studies [38–40]. The authors offer two reasons as to why the *Cr* condition achieved the best performance among the others. (1) The *Cr* channel takes out the effect of illumination [40] because the *Cr* channel is the red component (*R*), eliminating the luminance component (*Y*) (Equation (7)). Therefore, the Cr channel is more suitable for studying images under different exposures. (2) The *Cr* channel, a chrominance component for red [41,42], eliminates the disturbance of the green chrominance component for segmentation in near-color backgrounds [39].

The results of this study demonstrate that adding the *Cr* channel improved the ability of YOLOv5s model to recognize cucumber fruits. However, the precision only increases marginally from 83.7% to 85.19%. This is due to the fact that the information of the Cr channel is lost anyway throughout the formal training process as the network weight continues to converge to the RGB color space over hundreds of cycles. Additionally, the loss of feature information results from the depth of the network layers.

*4.2. The Future Research Focus*

Inputting images in *Cr* color space together with *RGB* images into the model for training improved the model precision. However, the training data type of the YOLOv5 model is three-channel images, such as *R*, *G* and *B*. Moreover, the direct addition of the *Cr* channel led to the incompatibility of the model. To solve this problem, the author proposes two directions to address this issue (Sections 4.2.1 and 4.2.2).

### 4.2.1. Multi-Channel Parallel Convolution Neural Network

Through up-sampling, down-sampling, and other feature fusion processes, feature maps with various inputs and sizes can be combined to extract more features [43]. The RGB image and the YCbCr image simultaneously enter the YOLOv5 model, which is configured with two input terminals. The feature fusion is performed following the preliminary feature extraction. The YOLOv5 network structure now includes a parallel convolutional neural network module to serve the goal of fusion training of RGB and YCbCr pictures.

Similarly, Yang et al. [44] input the upper half and the lower half of the human face into a parallel convolution neural network for training and learning in order to accurately place the faces' important features. By fusing a parallel neural network with a residual network, Wei et al. [45] created a neural network model for classifying leaves, and the maximum accuracy was 90.67%.

### 4.2.2. Modify Model Operation Dimension

The learning and training of the YOLOv5 model for three-channel images are determined by its network structure. From the perspective of modifying the network structure, if the tensor operation of the YOLOv5 model is changed to be compatible with the 4-channel images so that it can process the 4-channel images, then the image in the *Cr* channel can enter simultaneously with the RGB images. Thus far, no published study has used this strategy, suggesting a direction for further study.

## 5. Conclusions

Through the ReliefF weight analysis, it was proved that, among the twelve channels of four color spaces, the *Cr* channel had the highest weight in distinguishing cucumber fruit in a near-color background. By pre-training the YOLOv5s model with the *Cr* channel, the enhanced YOLOv5s model had a 1.49% increase in precision, an 8.03% increase in AP, and an 8% increase in mAP. The results of the ablation experiment also prove the effectiveness of the Cr channel in improving the recognition ability of other deep learning models including SSD, Faster R-CNN, and YOLOv5s/m/l/x, where the precision was increased by 1.51%, 3.09%, 1.49%, 3.15%, 0.63%, and 2.43%, respectively. The authors also discussed the reasons for the good performance of the Cr channel, further providing a theoretical basis for the effectiveness of this method. In the future, we will work on improving the precision of the enhanced YOLOv5s model by using the multi-channel parallel convolutional neural network approach or modifying the network structure to accommodate four or more channels.

**Author Contributions:** Conceptualization: N.W. and T.Q.; data curation: N.W.; formal analysis: N.W.; funding acquisition: L.L. and J.Y.; investigation: Y.X. and X.Z.; methodology: N.W.; project administration: J.Y. and L.L.; resources: H.Z., Y.Z. and N.W.; software: N.W.; supervision: J.Z. and T.Q.; validation: T.Q.; visualization: N.W.; writing—original draft: N.W.; writing—review and editing: T.Q., J.Z. and Y.Z. All authors have read and agreed to the published version of the manuscript.

**Funding:** This work was supported by Shanghai Agriculture Applied Technology Development Program, China (Grant No. G2022015) and Shanghai Science and Technology Committee Program (Grant No. 21N21900700). The funders had no role in study design, data collection and analysis, decision to publish, or preparation of the manuscript.

**Institutional Review Board Statement:** Not applicable.

**Data Availability Statement:** The data presented in this study are available on request from the corresponding author. The data are not publicly available due to their containing information that could compromise the privacy of research participants.

**Acknowledgments:** The authors would like to thank Shenglian LU for his insightful guidance and Dong Hu and Chao Ma for their meaningful discussion.

**Conflicts of Interest:** The authors declare no conflict of interest.

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
