# Peer review of "An Enhanced YOLOv5 Model for Greenhouse Cucumber Fruit Recognition Based on Color Space Features"

_agriculture, doi:10.3390/agriculture12101556_

Round 1
Reviewer 1 Report
An enhanced YOLOv5 model for greenhouse cucumber fruit recognition based on color space features.
MAIN COMMENTS
This paper discusses the potential of convolutional neural networks for indoor fruits detection. Approach and results are quite interesting, but the work need a major revision before publication.
A total of 438 original photos were collected: it is not clear the number of plants 438 different plants/fruits? or many photos repeated on the same plant? please specify. In case the number of plants/fruits is much lower than 438, then the authors should better clarify how results might be affected by larger uncertainty (due to not independent images).
Did the sample included different dimensions?
Why not including different varieties?
Please better describe the sample (maybe through a table): how many occluded fruits images? how many under/over-exposed images? how many images with two or more fruits?....
Equations 8, 9 and 10: too many digital numbers: three or four decimals might be n general sufficient.
It is not clear if photos have been taken also in the case of plants with no fruits. Such images might be useful in order to estimate false positives.
The relevance and the improved performance of Yolo v5 with respect to previous version must be discussed (maybe referring to previous works as e.g.: https://doi.org/10.3390/agronomy12020319).
Author might discuss the time needed by different analysed techniques.
The author should better explain and motivate in the paper why the Cr condition provided the best results.
I am not sure, but results are apparently not so good. Or better are they enough to allow some kind of automation operation in greenhouses (e.g.: https://doi.org/10.3390/agriculture12030346 or https://doi.org/10.3390/agriculture12060856)? Please discuss.
OTHER COMMENTS
Please revise text and English.
Remove/rewrite lines 217-219 "This section may be divided by subheadings. It should provide a concise and precise description of the experimental results, their interpretation, as well as the experimental conclusions that can be drawn."
Separate figure captions from the following paragraph.
The relevance of the names given to different cucumbers conditions in figure 2 is not clear: are these names (e.g.: “1Partialcucumber” or “2Slicedcucumber") relevant for the paper? if not they might be removed.
Figure 6: separate the 9 figures with a thin white space.
The size of figures 10 might be reduced a bit.
Lines 319-320 and other: put references at the end of sentences and not at the beginning.
If the paper is proposed for further revision, in the replay to thee referees, clearly highlight the sentences (e.g. with yellow colour) which have been corrected/added during revision.
Author Response
Point 1:A total of 438 original photos were collected: it is not clear the number of plants 438 different plants/fruits? or many photos repeated on the same plant? please specify. In case the number of plants/fruits is much lower than 438, then the authors should better clarify how results might be affected by larger uncertainty (due to not independent images).
Response: Thank you for your comments. In our experiment, there were 438 original photos were collected from 200 cucumber plants which were randomly selected from 720 cucumber plants population. In order to increase the dimensionality, 2 or 3 fruit images were taken at random orientations and exposures for every sample plant (Table 1 & Figure 1). Occlusions (Figure 2) were also considered to increase the randomness and diversity of training samples (Lines 132-138). Furthermore, we applied gaussian noise (Figure 3a), random noise (Figure 3b) and salt and pepper noise (Figure 3c) to extend the data set from 438 to 801, and labeled 2191 positive labels and 1752 negative labels (Lines 163-167). We believe that the image dataset is sufficient for model training.
Point 2:Did the sample included different dimensions?
Response: Thank you for your questions. As mentioned in the first response, we considered the dimensions of the dataset at three aspects: we set a variety of camera orientations (Figure 1), different light conditions, and different degrees of occlusion (Figure 2). To clear our experimental design, the specific data information was added in Table 1 (Lines 141).
Point 3:Why not including different varieties?
Response: Our experiment was conduct in a Venlo- type glass greenhouse. The European type cucumber like Delta star RZ F1 hybrid cucumber is the most common variety grown in this type of greenhouse. The European cucumber is a female type, with a large number of fruits and high labor costs for harvesting, so automated harvesting technology is urgently needed. On the other hand, the main purpose of this paper was to enhance the recognition precision of green cucumber fruit in the near-color background. For the above two reasons, we didn’t use different varieties to study. But in the future we are going to use other varieties to test the robustness of our model.
Point 4:Please better describe the sample (maybe through a table): how many occluded fruits images? how many under/over-exposed images? how many images with two or more fruits?....
Response: Thank you for your comments. All the photos contain more than two cucumber fruits. After data extension, the number of occluded fruits, under/over-exposed images is 326, 336 and 102 respectively. We supplemented the description of the dataset in Table 1(Line 141). We hoped the revised version is more clear in the experimental description.
Point 5:Equations 8, 9 and 10: too many digital numbers: three or four decimals might be n general sufficient.
Response: Thank you for your comments, the author had modified it.
Point 6:It is not clear if photos have been taken also in the case of plants with no fruits. Such images might be useful in order to estimate false positives.
Response: Thank you very much for your constructive comments. Our data collection was carried out during the fruiting period, and we were also realized the importance of negative sample for model training. Therefore, we have marked negative labels (Lines 157-159) in the data set, which can measure false positives to some extent (Figure 2).
Point 7:The relevance and the improved performance of Yolo v5 with respect to previous version must be discussed (maybe referring to previous works as e.g.: https://doi.org/10.3390/agronomy12020319)
Response: We are grateful for your thoughts in the review. In the earlier literature review, the author learned about the improvement of YOLOv5 and discussed it in the manuscript (Lines 214-232). As a result, the author did not put up a comparative experiment with earlier versions of YOLO due to the advancement of YOLOv5. Instead, the differences between the four versions of the YOLOv5 model, SSD and Faster R-CNN were compared (Table 4) in this paper. In the discussion, comparing with YOLOv3 and YOLOv4, we added the improvements of YOLOv5 and their role in this study. (Lines 437-443)
Point 8:Author might discuss the time needed by different analysed techniques.
Response: ‘‘Thank you for your suggestion. In the revised version, we used the FPS (Table 4) value to measure time consumption. The information about FPS has been supplemented to the article as a control data for ablation experiments. The frame rates for the SSD and Faster R-CNN models are 28.9 and 22.5. However, the precision of these two models were only about 55%. Besides, along with the network depth, the FPS of YOLOv5 continues to decline. From 53.3 FPS of s version to 14.7 FPS of x version. Because this study did not involve changes in the structure of the YOLOv5 model, the change in FPS before and after improvement was not large. (Lines 347-352)
Point 9:The author should better explain and motivate in the paper why the Cr condition provided the best results.
Response: Thank you for your professional comments. Through the studying of Cr channel characteristics and the analysis of the channel converting mechanism, and further combining our results, the authors offer two reasons for best performance of Cr condition among the others. 1) The Cr channel takes out the effect of illumination, because Cr channel is the red component(R) eliminating the luminance component(Y)(Eqn.7). Therefore, Cr channel is more suitable for studying images under different exposures. 2) The Cr channel, a chrominance component for red, eliminates the disturbance of the green chrominance component for segmentation in near-color background. (Lines 457-464)
Point 10:I am not sure, but results are apparently not so good. Or better are they enough to allow some kind of automation operation in greenhouses (e.g.: https://doi.org/10.3390/agriculture12030346 or https://doi.org/10.3390/agriculture12060856)? Please discuss.
Response: Thank you for your comments. The background of the images in our dataset is quite complicated. The background contains not only various facility components, but also non-target organs with similar colors, such as leaves, petioles, stems, etc. At the same time, the fruit target is usually obscured by other organs so that only part of them are visible. These complex backgrounds also greatly increase the difficulty of model identification. Although using a single-color background can significantly improve the model recognition precision, as shown in Kai Jiang’s study, it is difficult to achieve in practical applications in the greenhouses. In order to further improve the recognition precision, the author had discussed in Section 4.2 (Lines 474-503), which will be the author's next research step.
OTHER COMMENTS
Please revise text and English.
Point 11:Remove/rewrite lines 217-219 "This section may be divided by subheadings. It should provide a concise and precise description of the experimental results, their interpretation, as well as the experimental conclusions that can be drawn."
Response: Thank you for your comments, the author has rewritten the sentences.
Point 12:Separate figure captions from the following paragraph.
Response: Thank you for your correction of the manuscript format. The author has revised it as required.
Point 13:The relevance of the names given to different cucumbers conditions in figure 2 is not clear: are these names (e.g.: “1Partialcucumber” or “2Slicedcucumber") relevant for the paper? if not they might be removed.
Response: Thank you for your comments. The author had rewritten the part of classes explanation. We set positive and negative labels as two classes (Figure 2), and divides the positive labels into three categories (a), (b) and (c) according to the degree of occlusion of cucumber fruits, and takes cucumber flowers (d) and leaves (e) as negative labels (Lines 157-160). We hope that the standard of classification had been accurately described.
Point 14:Figure 6: separate the 9 figures with a thin white space.
Response: The author has modified Figure 6 according to your opinion, and the modified picture looks more concise. (Figure 6)
Point 15:The size of figures 10 might be reduced a bit.
Response: Thanks for your suggestion, the author changed Figures 9 and 10 to fit the layout's size, giving the manuscript a more attractive and readable appearance.
Point 16:Lines 319-320 and other: put references at the end of sentences and not at the beginning.
Response: Thank you very much for pointing out the mistakes in the manuscript format, and the author also checked the reference format of the full text.
Point 17:If the paper is proposed for further revision, in the replay to the referees, clearly highlight the sentences (e.g. with yellow color) which have been corrected/added during revision.
Response: Your suggestions for the revision process are really useful. All changes in the revised version are in track mode, and the newly added text has been highlighted.
Reviewer 2 Report
The manuscript requires major corrections before the next consideration:
An abstract is often presented separately from the article, so it must be able to stand alone. Hence the problem statement, aim, novelty and results of the study has all included in.
The abstract need to be rewritten
the research gap is lost in the introduction section
in the "Acquisition and Processing of Datasets" did you employed statistical analysis? the descriptions are not clear
you need to add statistical analysis such as ANOVA for analysing the dataset before the presentation of the modeling results.
in the modeling section the description for model the YOLO technique for the present study is not clear. how did you select the modeling parameters? How did you obtain the best model architecture?
the discussion section need to be validated with the conducted studies
the conclusion section need to be rewritten.
what is the future perspective?
please clearly describe the experimental validation
Author Response
Point 1:The manuscript requires major corrections before the next consideration:
Response: Thank you for your comments. We have made detailed revisions and modifications to the content. We have carefully supplemented and rewritten the abstract, discussion and conclusion and also make a more detailed supplementary description in Materials and methods.
Point 2:An abstract is often presented separately from the article, so it must be able to stand alone. Hence the problem statement, aim, novelty and results of the study has all included in.
Response: We appreciate your comments, and we have rewritten the abstract based on your suggestions. We believe that the revised abstract is now clearer and more informative.
Point 3:The abstract need to be rewritten.
Response: Thank you for your comments, we have rewritten the abstract part of the manuscript. (Lines 16-23, Lines 31-37 & Lines40-41)
Point 4:the research gap is lost in the introduction section.
Response: Thank you for your feedback on the Introduction section. We have added and filled the research gaps in the Introduction section. (Lines 107-113)
Point 5:in the "Acquisition and Processing of Datasets" did you employed statistical analysis? the descriptions are not clear.
Response: Yes, we used ANOVA on significance analysis. The analysis method and analysis results are supplemented in figure 7.
Point 6:you need to add statistical analysis such as ANOVA for analysing the dataset before the presentation of the modeling results.
Response: Possible differences in the weight between the channels were tested with an ANOVA. The analysis showed that there were significant differences between different channels. Differences between the channels were tested with a pairwise comparison Tukey Contrast in the ‘glht’ function in the ‘multicomp’ package of RStudio version 1.2.5033. And the result showed that Cr channel had significant differences (p < 0.001). (Figure 7)
Point 7:in the modeling section the description for model the YOLO technique for the present study is not clear. how did you select the modeling parameters? How did you obtain the best model architecture?
Response: Thank you for your questions about the manuscript. At the beginning of the study, we have learned the development history of the object detection model based on deep learning since 2012, and learned the problems solved after each iteration of the model and the defects that still exist. Considering the performance, and applicability, we chose the YOLOv5 model.
The deep learning network model can have tens of thousands or even more parameters. To manually set or train from scratch is unfeasible. To relief the training strain and time requirements, the model's parameters were taken from the COCO public data set. YOLOv5 have four different depth structures: s, x, m, and l. we selects the s structure with the least volume as the ideal model structure. This model could solve the issues of information loss, growing training time, and challenge of deployment by the deepening of model depth. More detailed discussion about comparison between YOLO v5 with other versions have been added to the Discussion section.
Point 8:the discussion section needed to be validated with the conducted studies.
Response: Thank you for your opinion, the author has rewritten the manuscript content of the discussion section. (Lines 429-432, Lines 437-451 & Lines 457-464)
Point 9:the conclusion section need to be rewritten.
Response: Thank you for your comments. The author rewritten the conclusion section. (Lines 505-509 & Lines 512-524)
Point 10:what is the future perspective?
Response: I sincerely appreciate your informative remarks on the manuscript's content. In order to further improve the model's capacity to recognize cucumber fruits, the author considers two aspects of data and network topology (Lines 481-503). Add channels (like Cr) to RGB visible light picture data to enhance the image information. We can also change the computation dimension and add new modules to the network structure to make the model run more quickly and accurately.
Point 11:Please clearly describe the experimental validation.
Response: Thank you for your comments, the author has rewritten the experimental verification section. (Lines 254-258)
Round 2
Reviewer 1 Report
I think the authors have done a good job to improve the quality of the paper.
Still English language might be further improved.
And still I believe the results at the end are not so good to allow automatic harvesting.
Nevertheless, I believe the work will be acceptable after English revision.
Author Response
Thank you for your professional comments. It helps us a lot to improve the quality of our manuscript. We have made further detailed modifications to the English. We believe the English expression of the revised version is clearer and better understood.
Reviewer 2 Report
All the comments have been successfully addressed.
Author Response
Thank you for your professional comments. It helps us a lot to improve the quality of our manuscript. We have further revised the English in detail. We believe the English expression of the revised version is clearer and better understood.